# Client-Centered Breastfeeding-Promotion Strategies: Q Methodology

**DOI:** 10.3390/ijerph18062955

**Published:** 2021-03-13

**Authors:** EunSeok Cha, Myoung Hwan Shin, Betty J. Braxter, In Sook Park, Hyesun Jang, Byung Hun Kang

**Affiliations:** 1College of Nursing, Chungnam National University, Daejeon 35015, Korea; ispark@cnu.ac.kr (I.S.P.); mitotoim@gmail.com (H.J.); 2Nell Hodgson Woodruff School of Nursing, Emory University, Atlanta, GA 30322, USA; 3 Industry-University Cooperation+, Kyungsung University, Busan 48434, Korea; 4School of Nursing, University of Pittsburgh, Pittsburgh, PA 15213, USA; bjbst32@pitt.edu; 5College of Medicine, Chungnam National University, Daejeon 35015, Korea; missinglime@cnuh.co.kr

**Keywords:** breastfeeding, breastfeeding psychology, breastfeeding trends, public-health practices, Korean women, Q methodology

## Abstract

Fewer Korean women are choosing the 6 months of exclusive breastfeeding that are recommended for obtaining its maximal benefits despite an increasing effort to promote breastfeeding. Successful breastfeeding education and counseling need to be segmentally designed on the basis of client characteristics. This study explored the perceptions of breastfeeding in pregnant and 6 month postpartum Korean women using the Q methodology, a useful research approach to examine personal perceptions, feelings, and values about a concept or phenomenon of interest and identify typologies of perspectives. The Q sample consisted of 38 statements representing the universe of viewpoints on breastfeeding. The P sample (N = 49) included women who shared their perceptions of breastfeeding and filled each grid with a statement in the Q sorting table. Data were analyzed using the PC-QUANL program. Varimax (orthogonal) rotation revealed four factors that explained 53.0% of variance: maternal privilege (Factor 1), option based on emotion (Factor 2), option if efficient (Factor 3), and option if I have sufficient problem-solving skills (Factor 4). Korean women have changed their attitudes toward breastfeeding, with all participants viewing breastfeeding as optional. Breastfeeding-promotion campaigns and education need to consider societal norms and changes in women’s beliefs and perceptions.

## 1. Introduction

Breastfeeding provides many physical and emotional benefits to babies and mothers [1,2,3]. Human milk is the optimal nutritional source for infants [1,4], as it helps to advance physical, neurological, and cognitive development. It decreases infant infections, allergies, and morbidity and mortality compared to artificial formula feeding [1,5]. For mothers, breastfeeding prevents various postpartum complications such as uterine atony, depression, and proinflammatory responses [6]. Additionally, there are many long-term benefits for the mother’s cardiometabolic health (e.g., diabetes, postpartum stroke) and sex-related cancers (e.g., breast and ovarian cancer) and the baby’s lifelong health [1,7]. On the basis of these findings, the United Nations Children’s Fund (UNICEF) recommended the early initiation of breastfeeding within the first hour of birth, and set a global target of 70% of 6 month exclusive breastfeeding (EBF) by 2030 [3,8].

Korean women show a unique trend with breastfeeding. More than 88% of women practice breastfeeding during the first 3 weeks after birth, but only 48.1% of women continue breastfeeding through 6 months [3,9]. The exclusive breastfeeding rate in the first week of birth is only 16.1%, but increases up to 40.4% by the first month of birth. Then, it decreases to 34.5% at 3 months, 30.5% at 4 months, and 14.9% at 6 months [9], which is much lower than the 6 month EBF in the United States (25.6%) and the global target rate (70%) by 2030 proposed by UNICEF [3,8,10,11].

A dramatic decline in breastfeeding practice in Korea has occurred during the past few decades [11,12,13]. To promote breastfeeding, diverse efforts led by the government are continuous, such as education during the antenatal period emphasizing the advantages of breastfeeding, postpartum care service with lactation massage, and a government voucher for a certified postnatal care helper through a public-health center for any woman in need. The Baby-Friendly Hospital Initiative (BFHI) and the Mother-Friendly Workplace Initiative (MFWI) were also implemented in hospitals and workplaces [14]. However, the number of women who chose the exclusive 6 month breastfeeding that is recommended for obtaining the maximal benefits of breastfeeding has declined over the years [9,13].

Previous research reports multiple factors contributing to the negative milieu toward initiating and continuing breastfeeding [15,16]. Barriers to breastfeeding on the policy level include an unsupportive policy of maternal and paternal leave before and after childbirth [12,15], the lack of adequate breastfeeding spaces in the workplace or in public sites that afford privacy [17], aggressive formula marketing by formula manufacturers, and an unsupportive environment for breastfeeding in healthcare facilities [18]. Obstacles on the sociocultural level include social norms against breastfeeding in public places [19,20] and incorrect beliefs toward colostrum [19]. On the individual level, issues related to limited problem-solving skills to deal with challenges (e.g., desire to follow usual diet, inconvenience of pumping at work, emotional distress, and insufficient knowledge of baby’s nutrition requirements). Infant or maternal lactation issues also contribute to early discontinuation [16,21]. Additionally, concerns about older children, method of delivery [12,22], and the absence of partner and/or family support negatively influence breastfeeding practice [20].

Women do not automatically initiate and continue breastfeeding [12,23]. Rather, they voluntarily select breastfeeding on the basis of physical conditions, personal values, and social and cultural norms [23,24]. Therefore, successful breastfeeding education and counseling need to be segmentally designed on the basis of the mother’s characteristics [25].

The Q methodology is a useful research methodology to explore human subjectivity related to one’s personal point of view about a concept or phenomenon of interest, and to identify typologies of perspectives [26,27,28]. Each participant (P sample) responds with their views on a Q sort table (Q sorts), and factor analysis is conducted with the collected Q sorts to find factors, a small number of sets of sorted statements that differ from each other but show similarity among respondents [27]. Standardized (Z) scores, a weighted average of the values that the Q sorts most closely related to a factor help researchers to classify factors and identify archetypical perspectives [26,27,28], which provides a clue for developing person-centered marketing strategies to change behaviors [28,29]. It also allows for researchers to identify the characteristics of highly influential persons providing a strong effect on others’ opinions and behaviors, which can be useful information for designing a public-health intervention using community-based participatory research or social marketing [27,29].

The purpose of this study was to examine current perceptions of South Korean women toward breastfeeding using the Q methodology [30]. The findings could facilitate a better understanding of why Korean women practice or refrain from breastfeeding. Ultimately, these findings could contribute to the development of successful client-centered breastfeeding-promotion initiatives in Korea.

## 2. Materials and Methods

### 2.1. Study Design

This study employed the Q methodology, an integration of quantitative and qualitative methods that was developed by William Stephenson in 1935 [26,31]. This semiqualitative methodology employs a semiquantitative technique to explore human perspectives and classify perspectives [26,27]. Because of this unique approach to research, an increasing number of social scientists applied the Q methodology to answer their research questions. According to a PubMed search using the keywords of “q method* or Q sort*”, 1386 results were generated from 1952 to 2021; the Q methodology has been more popular since 2015. In 2020 alone, 121 papers using Q methodology were cited in PubMed.

### 2.2. Research Setting

Participants were recruited from antenatal education programs at public-health centers, lactation support provider clinics, and an obstetrics and gynecology (OBGYN) clinic affiliated with a university in South Korea.

### 2.3. Research Procedure

The Q methodology requires the following six steps: (1) concourse preparation (a representation of the universe of viewpoints), (2) final statement selections from the concourse (Q sample), (3) selection of Q sorters (P sample), (4) Q sorting, (5) factor analysis, and (6) factor extraction and interpretation. The Q methodology, therefore, requires two sample types: Q and P samples (Figure 1). Of the six steps, concourse development, Q-sample selection, and factor interpretation employ a qualitative methodology, while factor analysis and factor extraction apply a quantitative technique. To select P samples (i.e., Q sorters), either a convenient-sampling or purposive-sampling method is used [31].

### 2.4. Q Sample

Concourse, the Q population, is a hypothetical concept that includes voluminous and communicable statements on a topic, concept, concern, or interest [27]. Ideally, the concourse includes all possible opinions, perspectives, and expressions around the topic of interest since statements in the Q methodology are self-referential (e.g., “I think”, “in my judgment”, “from my point of view”), and each person has different views on a specific topic [27,28]. From the concourse, Q samples were selected with an unstructured or structured sampling method on the basis of the use of explicit experimental design [28,31]. While there are no hard and fast rules to select the Q sample (i.e., Q set), a number of 40–80 statements is accepted as the standard. However, any number for a Q set is acceptable as long as the Q sample has comprehensive coverage, and is pragmatic in terms of time and effort [27,32].

In the current study, a concourse consisting of 172 statements was extracted as a result of interviews with three lactation support providers and two women with breastfeeding experience. Additionally, the scientific literature, published offline materials (books, newspapers, magazines), and online sources (blogs and websites) were reviewed to find comprehensive and universe statements related to breastfeeding [28,31].

The Q sample is a group of naturalistic and representative statements on a specific topic, herein, breastfeeding [26,32]. Since there were no existing statements related to Korean women’s perspectives on breastfeeding, an unstructured sampling method was applied with the consideration to cover the perspectives of the person, woman, mother, and partner (i.e., wife) domains. Positive, neutral, or negative points of view on breastfeeding were also included with a consideration of balancing negative and positive statements. Statements demonstrating a diverse context of breastfeeding (e.g., having other children, mother worked outside the home, an unsupportive partner on breastfeeding practice) were included. The first author and two Q methodology experts (2^nd^ and 4^th^ authors) had multiple discussions to select the Q sample, and 38 statements remained to represent the breastfeeding perspectives. Once the Q sample had been determined, a 9 range quasi-normal distribution table (Q sorting table) ranging from “strongly disagree” (−4) to “strongly agree” (+4) was generated (Figure 2).

### 2.5. P Sample

The P sample is participants who share their viewpoints on the topic of interest using a Q sorting table [26]. Unlike the usual quantitative research approaches, the Q methodology is not focused on generalizing to a population of people, but on finding the existence of particular viewpoints on a specific topic under particular circumstances [31,32]. Previous research recommends a P sample of 40–60 participants, while even a single case is acceptable depending on the research question [31,32].

In the current study, inclusion criteria were pregnant women or those who had given birth within the past 6 months. Exclusion criteria were not having a physician’s recommendation for breastfeeding due to a health condition and an inability to read Korean. A targeted sample of 50 women who had met the eligibility criteria were recruited using flyers and snowball sampling.

### 2.6. Data Collection

Data were collected between 1 July 2019 and 16 December 2019. Of 66 contacts, written informed consent was obtained from 50 participants (75.8%) prior to study implementation. Subsequently, participants were asked to complete each grid for each Q sample in the Q sorting table. To do this, each participant read the Q set consisting of 38 statements and sorted each statement into three categories: agree, disagree, and neutral. Then, the participants ranked the statements to cover the Q sorting grids. Once they had completed the placement of the statements, a trained research assistant questioned them regarding their rationale for the placements and recorded their responses. The Q sorts took 20–40 min to complete.

### 2.7. Data Analysis

The PC-QUANL program (version 1.2) was used to analyze the Q sorts [33]. Factor analysis was conducted to reveal patterns in the data after participants’ responses had been entered into the database and transformed into scores [26,34]. The statements posted on the left side with which participants disagreed were scored negatively, while the statements on the right with which participants agreed were scored positively.

For factor extraction, the eigenvalue (≥1), percentage of total variance by factor, the cumulative explained variance by all factors, and P-sample numbers in a factor were considered. With regard to the comparisons among factors, Z scores, standard scores in a normal distribution, were used to identify differences. There were associations between Z and p scores. If Z scores were 1.65, 1.96, or 2.58, p values were 0.10, 0.05, or 0.01, respectively. While absolute values were important, the directions (+/–) were meaningless in the comparison [31].

To label each factor, three independent researchers reviewed the arrays of the statements to capture the factor characteristics. Interview data from the P sample who had showed the highest Z scores in each factor also helped researchers to label factors. Labels for the factors were set once the three researchers with expertise in qualitative research or Q methodology had reached agreement.

### 2.8. Ethical Considerations

The institutional review board of the institution with which the principal investigator was affiliated approved the study (protocol code: 201904-SB-042-01; date of approval: 25 June 2019). All participants had given their written informed consent for inclusion before they participated in the study. Participants who had completed the Q sorts received a gift card of KRW 10,000 (approximately USD 10).

## 3. Results

### 3.1. Patient Characteristics

The final P sample contained the responses of 49 women. One participant provided her answers on multiple grids; thus, we excluded her responses. The participants were 24 to 41 years of age (mean age = 33.4 ± 3.49). All participants were married; 13 (26.50%) were primiparas; 17 (34.70%) reported no prior breastfeeding experience. Thirty participants (60%) had given birth in the past 6 months, and 19 (38.80%) were pregnant at the time of the study. The majority of the participants had college-level education or higher (see Table 1).

### 3.2. Profiles of Each Factor

Employing varimax (orthogonal) rotation, we identified four factors that explained 52.95% of the variance: Factor 1—maternal privilege (36.54%), Factor 2—option based on my emotional condition (8.80%), Factor 3—option if efficient (3.92%), and Factor 4—option if I have sufficient problem-solving skills (3.69%). The chosen eigenvalues were Factor 1 = 17.90, Factor 2 = 4.31, Factor 3 = 1.92, and Factor 4 = 1.80. Correlations among the factors were 0.37 to 0.67.

#### 3.2.1. Factor 1—Maternal Privilege

Table 2 displays descriptions of statements and Z scores. About half of the participants (n = 23, 46.94%) loaded on Factor 1, which was a favorable view of breastfeeding and a privilege of motherhood. The most favored statements were “Breastfeeding is something only I can do for my baby” (#2: Z = 2.02) and “Breastfeeding endears my child to me even further” (#19: Z = 1.91). The statements with which participants most disagreed were related to a negative body image, such as “Breastfeeding makes me sexually unattractive” (#16: Z = −1.67), “Breastfeeding makes me feel like a cow” (#9: Z = −1.65), and “I feel like an animal during pumping” (#27: Z = −1.32). This group of participants perceived breastfeeding as convenient (#18: Z = 1.50), time-saving (#17: Z = 1.28), and healthy (#29: Z = 1.12).

The most influential P sample participant in Factor 1 (Factor loading weight = 3.09) was a 33 year old woman who was recruited from a lactation support provider clinic. She reported a very good intimate relationship with her partner. Perceived financial condition was moderate. She was a college graduate, not pregnant now, and had breastfeeding experience.

#### 3.2.2. Factor 2—Option Based on My Emotional Condition

Thirteen or 26.53% of the participants loaded on Factor 2. These participants expressed that breastfeeding is an option and should always be performed when in a good mood. If breastfeeding causes stress (#4: Z = 2.17) or upset/annoyance (#24: Z = 1.54), an alternative needs to be considered. Breastfeeding in a public place caused embarrassment (#14: Z = 1.18). These perspectives align with the most disagreed-with statements and were placed on the left. These women neither considered breastfeeding a maternal responsibility (#1: Z = −2.03) nor a way to be a (perfect) woman (#15: Z = −1.95). Compared to the women who loaded on Factor 1, they had less breastfeeding experience, but had a similar mean age and pregnancy history.

The most influential P sample in Factor 2 (Factor loading weight = 1.46) was a 35 year old woman who had recently given birth (within the prior 6 weeks) and reported a medium–high intimate relationship with her partner. Perceived financial condition was moderate. She was a college graduate and had breastfeeding experience.

#### 3.2.3. Factor 3—Option if Efficient

Slightly more than one-fifth of the participants were loaded on this factor (n = 11, 22.45%). While participants acknowledged that breastfeeding affects their daily lives (#34: Z = 1.19), they viewed it as convenient (#18: Z = 1.50) and time-saving (#17: Z = 1.00). However, hey needed reassurance regarding the misconception that size and weight are indicators of a healthy baby. They expressed the concern, “If the baby does not gain weight during breastfeeding, I would lose confidence, thinking that it is not helping (#12: Z = 1.30)”. The participants loading on this factor wanted to continue breastfeeding for longer than 6 months (#11: Z = −1.36), a decision that they independently made (#38: Z = 1.77) rather than basing it on external factors (family or spouse’s opinion). As with the women loading on Factor 1, those loading on Factor 3 considered breastfeeding a maternal privilege (#1: Z = 1.01). However, they did not consider it as a way to achieve the status of the perfect woman (#15: Z = −1.25). The women loading on this factor had the lowest average age and highest educational level when considering the four factors.

The most influential P sample in Factor 3 (Factor loading weight = 1.07) was a 38 year old woman who had been recruited from a lactation support provider clinic. She reported a medium intimate relationship with the partner, and her perceived financial condition was moderate. She was a college graduate and had breastfeeding experience.

#### 3.2.4. Factor 4—Option if I Have Sufficient Problem-Solving Skills

Two participants with pregnancy and child-rearing experiences loading on Factor 4 expressed positive views of breastfeeding and considered it a maternal privilege (#2: Z = 1.17). Their expectations and concerns seemed realistic; encouragement and support from professionals were important for their decision making (#7: Z = 2.03; Table 3). Additionally, unlike the women loading on Factors 2 and 3, they exhibited a family-centered attitude (e.g., #27: I do not want to breastfeed if my family views it negatively (Z = 1.68); #25: Breastfeeding takes up too much of my energy, and thus it difficult to pay other family members an adequate amount of attention (Z = −1.36)).

The most influential P sample in Factor 4 (Factor loading weight = 0.84) was a 30 year old woman who had recently delivered. She was a high-school graduate and was recruited in the OBGYN clinic. She reported a medium–high intimate relationship with her partner and had breastfeeding experience. Her perceived financial condition was moderate.

### 3.3. Comparisons among Factors

#### 3.3.1. Factor 1 vs. Others

The women loading on Factor 1 believed that breastfeeding is more convenient than bottle-feeding is when they go out (# 18. Diff. = 1.47), helps them to be a perfect woman ((#15, Diff = 1.38), and that it should be continued even if milk production is insufficient (#32. Diff = 1.35).

Breastfeeding was not a cause of embarrassment compared to other factors (#14. Diff. = −1.26). While they acknowledged that breastfeeding affected their daily lives and physical freedom, similar to Factor 3, they considered it a time-saving and convenient way to nurture the baby. The women loading on Factor 1 wished to control negative emotions in order to continue breastfeeding, while those loading on Factor 2 seemed inclined to discontinue breastfeeding if they experienced negative emotions.

#### 3.3.2. Factor 2 vs. Others

Factor 2 was unique with regard to the identified characteristics, as the women loading on this factor seemed independent, self-interested, and emotion-focused. The major determinant of breastfeeding was their own emotions rather than any external factor (Internet, friends, family) or expectations (maternal responsibility; #4. Diff. = 1.40). Any barrier (breastmilk deficit, no place for breastfeeding in public, negative emotions, pain) was identified as a reason to discontinue breastfeeding.

#### 3.3.3. Factor 3 vs. Others

Factor 3 was associated with positive views of breastfeeding, and decisions were based on internal (self-efficacy, motivation) rather than external (subjective norms, social support) factors. These women pursued efficiency. They needed reassurance that the baby’s weight was not the only standard used to determine growth (#12. Diff. = 1.46).

#### 3.3.4. Factor 4 vs. Others

Factor 4 revolved around problem solving. For the women loading on this factor, encouragement from professionals was important in decision making (#7. Diff. = 1.55). They also showed a family-centered attitude (#28. Diff. = 2.20). They did not mind breastfeeding in front of their husbands.

### 3.4. Consensus Items across Factors and Average Z Scores

As shown it Table 3, we identified six consensus items across the factors (|Z| >1). There was agreement of four concourses across all factors: (1) breastfeeding is my own choice (#38. Z = 1.48); (2) I do not mind breastfeeding in front of my husband (Z = 1.21); (3) my emotional state transmits to the baby through breastmilk, consequently affecting the baby’s health (#33. Z = 1.10); and (4) the baby’s lactation schedule affects my daily routine (#34. Z = 1.07). The two most disagreed-with items across the four factors were: (1) I avoid breastfeeding because the required breast care is expensive (#36. Z = −1.23), and (2) Breastfeeding makes me feel like a cow (#9. Z = −1.18).

## 4. Discussion

Breastfeeding rates are diverse by nation, culture, and ethnic group despite the ongoing efforts of the World Health Organization and UNICEF [3,8,35]. Some consider breastfeeding to be “old-fashioned”, while others view it as normative childrearing behavior [12,36]. Depending on the social context, higher education has served to promote breastfeeding (e.g., Korean-American women) [12] while also serving as a barrier (e.g., South Korean women) [30], which was reinforced by the present findings. Tailoring intervention strategies in response to client characteristics and the social context are necessary to promote breastfeeding practice.

This study demonstrates the changed perspectives and societal norms regarding breastfeeding in South Korean women, and suggests new opportunities to promote breastfeeding in response to changes in personal perceptions and societal norms. For instance, in the 1970s, over 90% of Korean women practiced breastfeeding, as this was the accepted normative behavior to raising a baby [12,37]. More recently, an increasing number of Korean women have been choosing either to forgo breastfeeding entirely, or to breastfeed for only a few weeks or months instead of exclusively breastfeeding for the first 6 months despite public-health efforts [38]. Korea’s unique postpartum cultural beliefs and rituals possibly pose major obstacles in this regard; there is a cultural belief that postpartum mothers deserve rest to restore their energy. Thus, professional postpartum care service has emerged as a lucrative business. About 50%–70% of new mothers stay in a postpartum facility for 1–2 weeks after hospital discharge, and the newborn is cared for mainly by the staff [12]. Therefore, the rate of exclusive breastfeeding in the first week is only 16.1%, increases to 36.5% in the 2^nd^ week, and to 40.3% in the 3^rd^ week, according to 2018 statistics [9].

The negative impact from this cultural belief and social phenomenon, however, may thwart extensive and culturally adapted applications of public-health initiatives (e.g., Baby Friendly Hospital Initiative), “mother and newborn rooming in together 24 h a day in postpartum centers”. For instance, Korean-American women holding the same cultural beliefs on postpartum care showed a very positive attitude toward breastfeeding, possibly influenced by positive peer norms and social infrastructures toward breastfeeding in the United States, and the unavailability of private postpartum centers. Korean American women, physically distant from their mother (-in-law) who traditionally provides postpartum care, hired a private postpartum care specialist and received care at home [12]. Thus, mother and newborn stayed in the same room for the first several weeks with instrumental and emotional support from this specialist. Many women using this service expressed great satisfaction with regard to the technical advice and practical information for breastfeeding [12]. A modification of this approach might be helpful for women who load on Factors 1 and 4.

Women loading on Factor 2, and to some extent Factor 3, may benefit from persuasion. Given that these women sought evidence to defend and support their decision to breastfeed or not, providing a positive rationale for breastfeeding may need to be incorporated. A previous study reported that formula-feeding mothers often experienced internal conflict with regard to their choice of feeding [39], and desired affirmation that they could still be good mothers [21]. Our participants were concerned about transferring their negative emotions and physiologically unhealthy conditions to the infant through breastfeeding, which were reasons for avoiding it. Many women were also concerned that infant breastfeeding needs deprived them of sleep, leaving them tired and depressed [40,41]. Breastfeeding, however, can improve sleep quality, reduce sleep disturbance at night, and help to prevent depression [41,42]. Thus, healthcare professionals need to assess women’s concerns, unmet needs, and unrealistic expectations related to breastfeeding in order to instill positivity and reduce anxiety via appropriate counseling and support [43]. Favorable peer norms and a breastfeeding-friendly environment can also help to develop a positive breastfeeding attitude for Factor 2. Possible options for interventions to reduce anxiety and concerns related to first-time breastfeeding mothers include cognitive behavioral therapy, guided imagery and music therapy, narrative intervention, or the utilization of breastfeeding support groups such as breastfeeding blogs [21,44,45].

Human beings seek rationality; thus, efficacy and efficiency are important aspects of decision making [46]. Our Factor 3 women evaluated the efficiency and cost-effectiveness of breastfeeding before making a decision. An educational approach providing up-to-date information to increase the perceived benefits of breastfeeding may help women loading on Factor 3 to make a positive decision [29]. For instance, breastfeeding—including its positions and the attention to the infant—enables the formation of maternal sensitivity, contributing to the infant’s cognitive, linguistic, and psychological development, and to stronger mother–baby bonding [2,47]. Positive mother–infant interactions contribute to the development of parenting skills, and a positive maternal self-image and satisfaction [41,42,47]. Babies learn positive facial expressions (e.g., smiling) and responsiveness via mirroring, further endearing them to their mothers and others, which facilitates stronger intimate relationships with them [42]. This information can be beneficial for Factor 3 mothers who pursue rational reasons for breastfeeding and favorably consider bottle-feeding practice with human breastmilk in order to accurately calculate the input and output (I/O) with the amount of breastmilk.

Previous research showed that encouragement from one’s spouse or family facilitates breastfeeding [12,48], and the most influential P sample in Factor 1 revisited this evidence; the person reported a very good relationship with her partner and showed a strong desire to breastfeed. This finding shows that partner and family support are very important influences on breastfeeding. Thus, couple- and family-centered interventions may be helpful for those persons such as those in Factor 4 who have a family-centered attitude. Additionally, positive messages, support, and coaching from healthcare professionals to improve problem-solving skills and self-efficacy might work for this group. About 75% of new mothers experience at least two breastfeeding problems (e.g., breast engorgement) that can lead to early weaning and postpartum depression in the first 2 months [21,49,50]. Practical skills and self-efficacy in breastfeeding can be improved with the timely support from lactation support providers and empathetic professionals [12,21,51], and verbal encouragement and emotional support from partners and family [49,52]. In addition, how to reduce the negative impact on the firstborn child (e.g., regression, emotional instability) when breastfeeding the second needs to be considered since it can be a major barrier for breastfeeding the second one [12].

Despite the significance of the study, there were limitations. About 53% of the variance was explained by four factors, leaving 47% unexplained. However, the purpose of the Q methodology is not to increase the explanatory power or identify frequency, but to effectively and efficiently capture diverse views uncovered within a population, and to understand the multiple perspectives [26]. Likewise, the eigenvalue or variance is a relatively meaningless concept in the Q methodology since it is excessively affected by the number of participants [53]. Another limitation was related to our P sample consisting of one ethnic group, and only pregnant and 6 month postpartum women who were highly educated. However, Koreans are considered to be one ethnic group, and more than two-thirds of childbearing Korean women are college graduates [11], meaning that it should not be a concern that high education in the P sample generated biased or skewed findings. Since intentions and behaviors of human beings are influenced by interactions with others and social context [54,55], future studies need to investigate the perceptions of breastfeeding within different populations (e.g., adolescent mothers or women experiencing high-risk pregnancies). Additionally, findings from cross-cultural research in Korean women of childbearing age who had immigrated from or emigrated to other countries would be helpful in designing a culture-specific intervention.

## 5. Conclusions

Societal norms on breastfeeding have changed over the past few decades, with Korean women now viewing breastfeeding as an option for feeding an infant rather than normative behavior. While this study showed that the majority of women have positive attitudes toward breastfeeding, the need was evidenced for positive encouragement and practical support from their partners, families, and healthcare professionals enabling women to be more empowered. Additionally, this study suggests a new public-health initiative with segmentally tailoring messages for women, partners, families, communities, and policy makers. Healthcare providers also need to be guided to deliver tailored counseling and education on the basis of patient characteristics and values.

## Figures and Tables

**Figure 1 ijerph-18-02955-f001:**
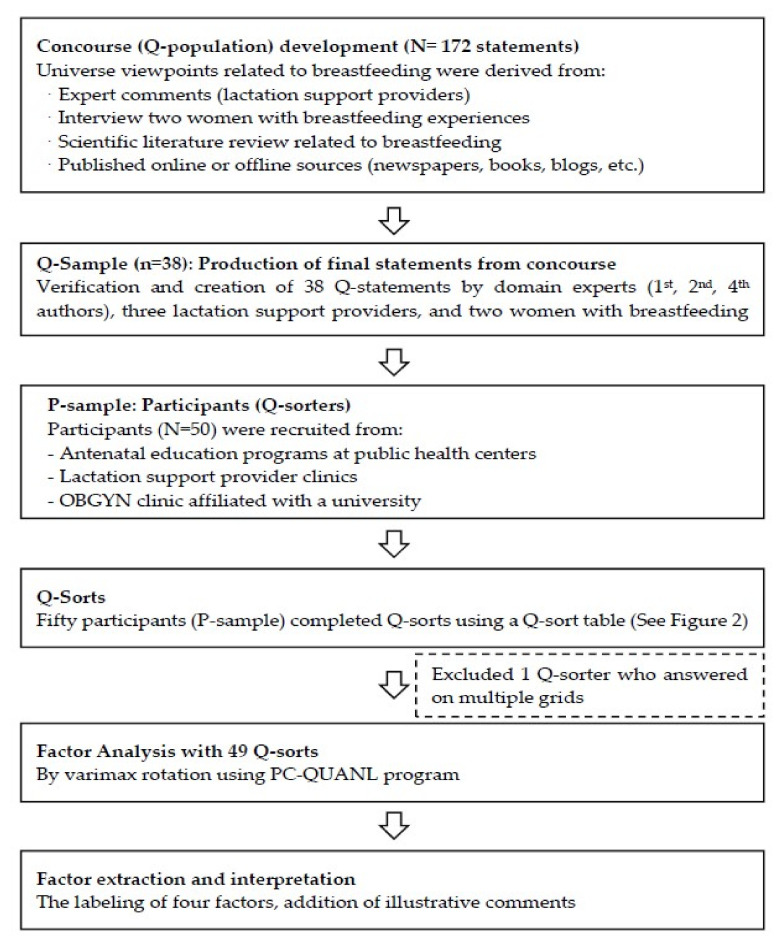
Q-methodology steps in current study.

**Figure 2 ijerph-18-02955-f002:**
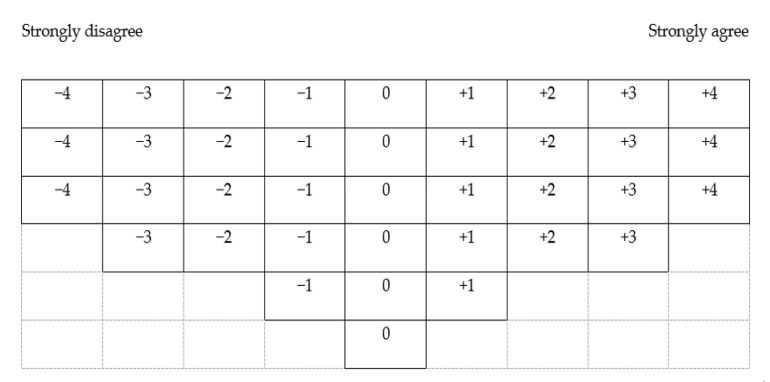
Q sorting table for Q sample (38 statements).

**Table 1 ijerph-18-02955-t001:** Eigenvalues, variance, and participant characteristics.

Variable	Factor 1 (*n* = 23)	Factor 2 (*n* = 13)	Factor 3 (*n* = 11)	Factor 4 (*n* = 2)	Total Sample (N = 49)
Eigenvalues (% of total variance)	17.90 (36.5%)	4.31 (8.8%)	1.92 (3.9%)	1.81 (3.7%)	N/A
Age (years), *n* (%)					
20–30	3 (13.0%)	2 (15.4%)	3 (27.3%)	1 (50.0%)	9 (18.4%)
31–40	19 (82.6%)	11 (84.6%)	7 (63.6%)	1 (50.0%)	38 (77.6%)
≥ 41	1 (4.3%)	0 (0.0%)	1 (9.1%)	0 (0.0%)	2 (4.0%)
Mean age ± SD	33.8 ± 0.49	33.4 ± 3.52	32.7 ± 4.81	33.0 ± 4.24	33.4 ± 3.49
Pregnancy history					
Yes	16 (69.6%)	9 (69.2%)	9 (81.8%)	2 (100%)	36 (73.5%)
No	7 (30.4%)	4 (30.8%)	2 (18.2%)	0 (0.0%)	13 (26.5%)
Breastfeeding history					
Yes	14 (60.9%)	7 (53.8%)	9 (81.8%)	2 (100%)	32 (65.3%)
No	9 (39.1%)	6 (46.2%)	2 (18.2%)	0 (0.0%)	17 (34.7%)
Currently pregnant					
Yes	8 (34.8%)	6 (46.2%)	5 (45.5%)	0 (0.0%)	19 (38.8%)
No	15 (65.2%)	7 (53.8%)	6 (54.5%)	2 (100%)	30 (61.2%)
Marital status					
Married	23 (100%)	13 (100%)	11(100%)	2 (100%)	49 (100.0%)
Unmarried	0	0	0	0	0 (0.0%)
Education					
Completed high school	5 (21.7%)	1 (7.7%)	1 (9.1%)	1 (50%)	8 (16.3%)
Completed college or university	16 (69.6%)	11 (84.6%)	10 (90.9%)	1 (50%)	38 (77.6%)
Completed grad program	2 (8.7%)	1 (7.7%)	0	0	3 (6.1%)

Note. SD: standard deviation.

**Table 2 ijerph-18-02955-t002:** Item descriptions with Q-set numbers and typical array Z scores.

Item	Z Scores			
	Factor 1	Factor 2	Factor 3	Factor 4
1. Breastfeeding is a woman’s responsibility.	0.11	**−2.03**	−0.62	0.32
2. Breastfeeding is something only I can do for my baby.	**2.02**	**1.03**	**1.01**	**1.17**
3. It is difficult for me to adhere to the baby’s breastfeeding schedule.	−0.03	0.34	0.77	**1.17**
4. If it causes stress, I’d rather not breastfeed.	0.50	**2.17**	0.95	0.85
5. If it causes pain, I’d rather not breastfeed.	−0.27	0.50	0.02	−0.02
6. Although the doctor has prescribed me to take medicines, I am worried that they will affect my baby through breastfeeding.	0.96	0.93	0.93	0.90
7. Healthcare professionals’ opinions on breastfeeding affect my decision making.	0.67	0.39	0.37	**2.03**
8. Information related to breastfeeding from the Internet, blogs, friends, and peers affects my decision making.	−0.12	0.55	−0.82	**−1.52**
9. Breastfeeding makes me feel like a cow.	**−1.65**	−0.79	**−1.29**	−0.99
10. I do not mind breastfeeding in front of my husband.	**1.31**	0.84	**1.31**	**1.38**
11. I will stop breastfeeding after 6 months because it does not help with the baby’s growth.	−0.92	0.30	**−1.36**	**−1.71**
12. If the baby does not gain weight during breastfeeding, I would lose confidence, thinking that it is not helping.	0.37	0.18	**1.30**	**−1.04**
13. Breastfeeding restricts my physical freedom.	**−1.49**	−0.48	0.79	0.19
14. Putting out my breast to breastfeed in public is embarrassing.	−0.38	**1.18**	0.96	0.51
15. I would become the perfect woman through breastfeeding.	0.03	**−1.95**	**−1.25**	−0.85
16. Breastfeeding makes me sexually unattractive.	**−1.67**	**−1.77**	**−1.43**	**1.36**
17. Breastfeeding is time-saving compared to bottle-feeding.	**1.28**	−0.88	**1.00**	0.19
18. When I go out, breastfeeding is more convenient than bottle-feeding.	**1.50**	−0.93	**1.50**	−0.48
19. Breastfeeding endears my child to me even further.	**1.91**	0.59	0.62	**1.04**
20. Breastfeeding brings me closer to my husband.	0.54	0.04	**−1.41**	−0.85
21. I avoid breastfeeding since it restricts medical treatment.	−0.86	−0.39	−0.90	−0.90
22. I avoid breastfeeding since it puts restrictions on my diet.	−0.61	−0.50	−0.11	−0.32
23. I am worried that increasing my food intake to facilitate breastfeeding would make me obese.	−0.24	0.07	−0.82	0.48
24. I’d rather bottle-feed my baby instead of turning my back on him/her because breastfeeding upsets/annoys me.	0.15	**1.54**	0.10	−0.35
25. Breastfeeding takes up too much of my energy, and thus it difficult to pay other family members an adequate amount of attention.	−0.65	−0.09	0.18	**−1.36**
26. I consider family members’ opinions when I make breastfeeding decisions.	−0.89	**−1.10**	**−1.65**	−0.64
27. The pumping makes me feel like an animal.	**−1.32**	−0.77	−0.86	−0.53
28. I do not want to breastfeed if my family views it negatively.	−0.68	0.37	**−1.27**	**1.68**
29. Breastfeeding is good for my health.	**1.12**	0.03	0.08	0.35
30. Bottle-feeding with breast milk is better than direct breastfeeding since it helps to check the amount of milk the baby has consumed.	−0.91	0.05	−0.49	−0.51
31. If the amount of colostrum is small, I lose confidence about breastfeeding.	−0.32	−0.31	0.81	0.64
32. Breastfeeding should be continued despite the deficiency of breastmilk.	0.81	**−1.15**	0.54	**−1.01**
33. My emotional state during breastfeeding affects the baby’s health.	**1.47**	**1.12**	**1.29**	0.51
34. The baby’s lactation schedule affects my daily routine.	**1.19**	**1.35**	**1.19**	0.56
35. If I breastfeed, family members should endure various inconvenience such as my unavailability to attend to the need of other family members.	−0.72	0.13	0.21	−0.35
36. I avoid breastfeeding since it costs a lot of money for breast care during breastfeeding.	−0.89	**−1.12**	**−1.21**	**−1.68**
37. The shape of my nipple affects my decision regarding breastfeeding.	0.11	0.26	0.43	**1.54**
38. Continuation of breastfeeding depends on my will.	**1.16**	**1.97**	**1.77**	**1.01**

Numbers with bold refer to |Z| ≥ 1.

**Table 3 ijerph-18-02955-t003:** Consensus items and average Z score (≥1).

Item	Z Score
38. Whether to continue breastfeeding relies on my will.	1.48
33. During breastfeeding, my emotional state affects the baby’s health.	1.10
34. The baby’s lactation habit affects my daily life.	1.07
9. I feel like a cow when I breastfeed my baby.	−1.18
10. I do not mind breastfeeding in front of my husband.	1.21
36. I avoid breastfeeding since it costs a lot of money for breast care during breastfeeding.	−1.23

## Data Availability

The data presented in this study are available on request to the first and corresponding author. The data are not publicly available due to ethical issue.

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
