# Peer review of "Client-Centered Breastfeeding-Promotion Strategies: Q Methodology"

_ijerph, 2021, doi:10.3390/ijerph18062955_

Round 1

Reviewer 1 Report

Thanks for the author’s efforts in improving this paper. However, I don't think the author has addressed my major concern on the rigor of this study. For example, I concern about the comprehensive and presentative Q-sample. The author claimed that “the first author and two Q methodology experts 127 (2nd and 4th authors) had multiple discussions to select the Q-sample, and finally, they included 38 items in the Q-sample.” This statement did not tell the reader what criteria are used to determine the inclusion and exclusion of items. Also, the author did not address my concern on the criteria for factor selection. Factor 4 only contains 2 participants, which makes me worry about the rationale of factor number. How do the authors determine the number of factors? What criteria the author used in this study? 

Author Response

Point1: I don't think the author has addressed my major concern on the rigor of this study. For example, I concern about the comprehensive and presentative Q-sample. The author claimed that “the first author and two Q methodology experts 127 (2nd and 4th authors) had multiple discussions to select the Q-sample, and finally, they included 38 items in the Q-sample.” This statement did not tell the reader what criteria are used to determine the inclusion and exclusion of items. Also, the author did not address my concern on the criteria for factor selection. Factor 4 only contains 2 participants, which makes me worry about the rationale of factor number. How do the authors determine the number of factors? What criteria the author used in this study? 

Response 1: Thank you for sharing your concern. Q-methodology is an integration of quantitative and qualitative methods to explore human perspectives and to classify perspectives. This approach employs qualitative methodology at the stages of concourse development, Q-sample section, and factor interpretation including factor labeling. On the contrary, quantitative technique, factor analysis, is used for factor extraction. To obtain the rigor of the research, we created the research team with the experts in Q-methodology, Qualitative research and Quantitative (i.e., factor analysis) research.

For factor extraction, eigen value, explained variance by factor, cumulative explained variance and P-sample numbers in a factor are considered like research in factor analysis. For instance, Factor 4, which has two Q-sorters, is the result of factor analysis without any manipulation by researchers. We included more detailed explanations in the method section ( See. 2.7. Data analysis).

Reviewer 2 Report

Dear Authors,

I have reviewed the revised version of your manuscript, which is substantially improved from the version you submitted originally. On that basis, I will recommend that the manuscript is accepted for publication. Congratulations on producing a excellent paper.

Kind regards,

The Reviewer

Author Response

Point2: I have reviewed the revised version of your manuscript, which is substantially improved from the version you submitted originally. On that basis, I will recommend that the manuscript is accepted for publication. Congratulations on producing an excellent paper.

Response 2: Thank you for your acknowledgement. The authors carefully reviewed English in accordance with your suggestion as well.

Reviewer 3 Report

Dear Authors
Congratulations on this work. I think you have made a great effort in modifying the paper and it is reflected in the high quality of the content. I would like to make a couple of suggestions in order to improve the discussion section. 
Line 331. You discuss public health initiatives. Are these initiatives aligned with the Baby Friendly Hospital Initiative (BFHI) promoted by WHO and UNICEF? That would be a good point to make.

Line 361. The literature citations used do not support the use of blogs as breastfeeding support techniques. Much literature on peer support in breastfeeding support groups exists. I recommend adding some specific citations.

Although it is not my decision, the withdrawal of an author (being an author of correspondence) is very striking. I think it is convenient to justify (to the editor) this decision because of the possible future problems that this decision may entail. 

I wish them every success and good luck with their work.

Author Response

Reviewer 3: I think you have made a great effort in modifying the paper and it is reflected in the high quality of the content. I would like to make a couple of suggestions in order to improve the discussion section.

Response 3: Thank you for your insightful comments and feedback.  

Point 4: Line 331. You discuss public health initiatives. Are these initiatives aligned with the Baby Friendly Hospital Initiative (BFHI) promoted by WHO and UNICEF? That would be a good point to make.

Response 4 : We believe that BFHI ,which is an excellent exemplar of public health initiatives, needs to be tailored based on cultural context. For instance, the BFHI proposes “Enable mothers and their infants to remain together and to practice rooming-in 24 hours a day in the hospital” as a key clinical practice. We would suggest to modify this statement to “mother and newborn rooming-in together 24 hours a day in postpartum centers’ for Koreans as you can see in line 361.

Point 5: Line 361. The literature citations used do not support the use of blogs as breastfeeding support techniques. Much literature on peer support in breastfeeding support groups exists. I recommend adding some specific citations.

Response 5: Thank you for your suggestion. We added one more citation related to peer support (see Sayres and Visentin, 2018).

Point 6: Although it is not my decision, the withdrawal of an author (being an author of correspondence) is very striking. I think it is convenient to justify (to the editor) this decision because of the possible future problems that this decision may entail. 

Response 6: The former author provided the agreement of changing authorship to the journal. She recently changed the institution and was unable to take responsibilities as author due to her personal situations.